# Dietary *Lactobacillus reuteri* SL001 Improves Growth Performance, Health-Related Parameters, Intestinal Morphology and Microbiota of Broiler Chickens

**DOI:** 10.3390/ani13101690

**Published:** 2023-05-19

**Authors:** Chunli Chai, Yaowen Guo, Taha Mohamed, Gifty Z. Bumbie, Yan Wang, Xiaojing Zeng, Jinghua Zhao, Huamao Du, Zhiru Tang, Yetong Xu, Weizhong Sun

**Affiliations:** 1State Key Laboratory of Silkworm Genome Biology, Key Laboratory of Sericultural Biology and Genetic Breeding, Ministry of Agriculture, College of Sericulture, Textile and Biomass Sciences, Southwest University, Chongqing 400715, China; chail@swu.edu.cn (C.C.);; 2College of Animal Science and Technology, Southwest University, Chongqing 400715, China

**Keywords:** *Lactobacillus reuteri* SL001, broiler, growth performance, serum biochemistry, intestinal morphology, gut microbiota

## Abstract

**Simple Summary:**

Recent research studies have revealed that probiotics such as *Lactobacillus* are good antibiotic alternatives in the poultry industry. We previously isolated *Lactobacillus reuteri* SL001 from the gastric contents of rabbits and proved that dietary inclusion of SL001 could positively improve the composition of the intestinal bacterial community in Alzheimer’s disease model mice. In the present study, we explored the effects of dietary SL001 on growth performance, health-related parameters, intestinal morphology and microbiota of broiler chickens. Our results showed that SL001 supplementation in diets promoted the growth performance of broilers, strengthened immunity, and improved antioxidant stress as well as intestinal morphology and microbiota, implying its potential application in boiler feeding.

**Abstract:**

It was assumed that dietary inclusion of *Lactobacillus reuteri* SL001 isolated from the gastric contents of rabbits could act as an alternative to feed antibiotics to improve the growth performance of broiler chickens. We randomly assigned 360 one-day-old AA white-feathered chicks in three treatments: basal diet (control), basal diet plus zinc bacitracin (antibiotic), and basal diet plus *L. reuteri* SL001 (SL001) treatment. The results showed the total BW gain and average daily gain (ADG) of broilers in SL001 treatment increased significantly (*p* < 0.05, respectively) compared with the control group from day 0 to 42. Moreover, we observed higher levels of immune globulins in both the SL001 group and the antibiotic group. Total antioxidant capacity and levels of antioxidant factors were also significantly increased (*p* ≤ 0.05, respectively) in the SL001 treatment group, while the interleukin 6, interleukin 4, creatinine, uric acid, total cholesterol, triglyceride, VLDL, LDL and malondialdehyde were remarkably decreased (*p* < 0.05, respectively). In the ileum of SL001 treatment broilers, the height of villi and the ratio of villi height to crypt depth were significantly increased (*p* < 0.05). Meanwhile, the crypt depth reduced (*p* < 0.01) and the ratio of villi height to crypt depth increased (*p* < 0.05) in the jejunum compared to the control. The abundance of microbiota increased in the gut of broilers supplemented with SL001. Dietary SL001 significantly increased the relative abundance of Actinobacteria in the cecal contents of broilers (*p* < 0.01) at the phylum level. In conclusion, *L. reuteri* SL001 supplementation promotes the growth performance of broiler chickens and exhibits the potential application value in the industry of broiler feeding.

## 1. Introduction

Broiler chickens are a vital source of high-protein meat for human consumption. Antibiotics have been extensively used in broilers to improve production performance, stimulate growth, and reduce morbidity and mortality. However, antibiotics can alter the normal intestinal microbiota of chickens, predisposing them to various diseases [1,2]. Moreover, the widespread use of antibiotics results in the emergence of antibiotic-resistant bacteria and drug residues in broiler products [3]. Therefore, there is currently an irresistible trend for the restriction or banning of antibiotic use in the feed industry [4]. In that case, the search for antibiotic substitutes is vital and has been popularized in research on feed. 

Many microbial communities exist in the intestinal tract of animals, which is crucial in the digestion and absorption of nutrients, immunity, and host defense against diseases [5,6]. Based on reports from more recent studies and clinical cases, increasing the number of beneficial bacteria in the intestinal tract impedes the growth of harmful bacteria, improves the diversity of intestinal flora, and thereby the body’s immunity [7,8,9,10]. Unlike antibiotics, probiotics cannot induce the emergence of drug-resistant bacteria or drug residues and are environmentally friendly [11]. Therefore, probiotics are ideal antibiotic substitutes to promote the gut health of birds. Probiotics used in the poultry industry are mainly *Lactobacillus*, *Bifidobacterium*, *Bacillus*, yeast, etc. [12]. As a member of normal flora in the animal gut, *Lactobacillus* produces bacteriocins that have bactericidal effects and acids that lower the gut pH, and competitively inhibit pathogens [13,14]. In a previous report, *Lactobacillus plantarum* P-8 was shown to activate beneficial immunological responses in broiler chickens and exert anti-inflammatory effects [15]. *Lactobacillus acidophilus* HM1, *Lactobacillus fermentum* HM3, and *Lactobacillus buchneri* FD2 showed significant inhibitory effects on three cancer cell lines (HT-29, MDA-MB-231, and HepG2) [16].

Previously, we screened and identified an *L. reuteri* strain named SL001 from the gastric contents of rabbits (NCBI login number: KU255544.1; species preservation number: CCTCC M 2019122) by using cholesterol as the only carbon source. Our previous animal studies revealed that SL001 has no toxic side effects on mice, and could improve mice’s intestinal microbiome [17]. The hypothesis that this strain has potential value and application in broiler production was deserved to confirm experimentally. Therefore, in the present study, we investigated the potential beneficial effects of in-feed *L. reuteri* SL001 on growth performance and carcass traits by comparing it with a growth-promotion antibiotic. Meanwhile, we also analyzed the serum biochemical parameters and the gene expression in the intestine to show the potential of this strain to promote the health status of broiler chickens. Additionally, the intestinal morphology and microbiome were also examined to investigate the possible action mechanism of this strain.

## 2. Materials and Methods

### 2.1. Experimental Strain

*L. reuteri* SL001 stain (NCBI login number KU255544.1; species preservation number CCTCC M 2019122) was isolated from the gastric contents of rabbits and reserved after characterization [17]. Before use, the strain stock was inoculated onto the Man Rogosa Sharpe (MRS) Medium plate, and incubated in an incubator at 37 °C for 24 h. A single colony on the MRS plate was inoculated into the MRS broth medium, then incubated in a shaking incubator at 37 °C for 18 h. After centrifugation at 4 °C for 15 min at 3000× *g*, the pellet was washed three times with sterile PBS buffer (pH 7.2). Then, the bacteria pellet was resuspended with a suitable amount of sterile PBS buffer. Finally, the bacteria suspension was thoroughly mixed with the basal diet to a final concentration of 3 × 10^8^ CFU/g.

### 2.2. Study Design and Animal Management

Arbor acre (AA) white-feather broilers and feeding diets were provided by Shide Poultry Co., Ltd. (Chongqing, China). Using a standard randomization procedure, 360 one-day-old AA white-feathered chicks were randomly assigned 20 birds/replicate, 6 replicates/treatment. There were 3 treatments in total, the basal diet (control), the basal diet with zinc bacitracin (antibiotic), and the basal diet with *L. reuteri* SL001 (SL001) treatment. The composition and nutrients of the basic diet are provided in Table 1. The control group was fed on the basal diet, whereas the antibiotic group was fed 0.5 g of a growth promotion antibiotic, zinc bacitracin (Mibbo, Shanghai, China), per kg of the basal diet. In the SL001 treatment group, chickens were fed a basal diet containing 3 × 10^8^ CFU/g SL001 bacterial suspension. The experimental duration was 42 days.

The experimental chickens were housed in suspended cages (60 cm × 50 cm × 30 cm) placed in an environmentally controlled room and tap water was available ad libitum via an automatic watering system. The room temperature was maintained at 33–35 °C for 1–2 days, then lowered by 2–3 °C every week and maintained at 20–25 °C in the fifth and sixth weeks, with a relative humidity of 30 to 70%. Temperature and humidity were continuously monitored and recorded. The animal room had continuous artificial lighting (12-h lighting, 12-h dark) and natural ventilation.

### 2.3. Growth Performance and Carcass Traits

The body weight (BW) of broilers was weighed once a week after fasting for 12 h and the feed intake of broilers was recorded to calculate the average daily gain (ADG), average daily feed intake (ADFI), and ADFI/ADG (F/G). 

After 42 days, 12 healthy broiler chickens were randomly selected from each treatment group, weighed, and sacrificed. Measurement of carcass weight (with the removal of blood, feathers, and internal organs, only the trunk, lungs, and kidneys are to be weighed) was performed, and the carcass percentage ((carcass weight/live weight) × 100%) was calculated. The breast, liver, gizzard, abdominal fat, bursa, spleen, heart, stomach, and pancreas from each chicken were weighed and recorded, and the percentage of organ weight to live body weight was calculated.

### 2.4. Measurements of Health-Related Parameters

Serum of 42-day broilers was separated by centrifugation at 3000× *g* at 4 °C for 10 min after collection and agglutination at 4 °C, and the serum samples were subjected to analysis for the health-related parameters using commercially available ELISA assay kits (Nanjing Jiancheng Biological Institute, Nanjing, China), which include immunological parameters: immunoglobulin A (IgA), immunoglobulin M (IgM), immunoglobulin G (IgG), complement 3 (C3) and complement 4 (C4), secretory immunoglobulin A (sIgA), interleukin 6 (IL-6), interleukin 4 (IL-4), interleukin 10 (IL-10), tumor necrosis factor α (TNF-α) and transforming growth factor β (TGF-β); antioxidant factors: malondialdehyde (MDA), superoxide dismutase (SOD), reduced glutathione (GSH), catalase (CAT), and total antioxidant capacity (T-AOC); and serum biochemical parameters: alkaline phosphatase (AKP), pepsin (PEP), lipase (LPS), amylase (AMS), glutamic-pyruvic transaminase (GPT), and glutamic oxalacetic transaminase (GOT), triglyceride (TG), total cholesterol (TCHO), low-density lipoprotein (LDL) and very low-density lipoprotein (VLDL), high-density lipoprotein (HDL), albumin (ALB), total protein (TP), creatinine (Cr) and uric acid (UA).

### 2.5. Intestinal Morphology

Six animals in each treatment group were randomly selected for slaughter on day 42. The mid-jejunum and mid-ileum segments were excised and rinsed with cold 0.9% saline, then fixed immediately into a 10% buffered formalin solution. Paraffin embedding, sectioning (HM325, Thermo Scientific, Waltham, MA, USA), and staining in hematoxylin and eosin solution were performed conventionally. The data of villi height and crypt depth based on the images of the villi were acquired using a Motic microscope with the software Motic Images 3.2 (BA410E, Motic, Xiamen, China). 

### 2.6. Total RNA Isolation and Quantitative Real-Time PCR Analysis

The total RNA of the liver and jejunum tissues were extracted, respectively, using the Trizol Reagent (15596026, Invitrogen, Carlsbad, USA) according to the manufacturer’s protocol. Then, cDNA was prepared using M-MLV reverse transcriptase (M1705, Promega, Madison, WI, USA), 5XM-MLV Buffer, Rnasin inhibitor (N2115, Promega, Madison, WI, USA), dNTPs (4030Q, Takara, Kyoto, Japan), and OligodT18 (5′-TTTTTTTTTTTTTTTTTT-3′, BGI Inc., Beijing, China). The cycling conditions for reverse transcription were 42 °C for 2 h, followed by 70 °C for 5 min. The cDNA samples were aliquoted and stored at −20 °C before use.

Quantitative real-time PCR (qRT-PCR) was performed using the ChamQ Universal SYBR Qpcr Master Mix (Q711-02, Vazyme, Nanjing, China) on a qTOWER 3G Real-Time PCR System (Lead Scientific Technology, Chengdu, China). The primers used for amplification in this study were provided in Table 2. Each amplification of qRT-PCR was performed according to the instructions recommended by the manufacturer. The housekeeping gene *β-actin* was selected as the internal control to normalize the expression of the other target genes. The relative gene expression levels were analyzed using the 2^−∆∆Ct^ method [18].

### 2.7. 16S rDNA Sequencing and Data Analysis

Cecal contents from six animals in each treatment group were sampled at d 42, and bacterial DNA was extracted using the E.Z.N.A.^®^ Stool DNA Kit (D4015, Omega, Inc., Norcross, GA, USA) according to manufacturer’s instructions. The forward and reverse primers (5′-CCTACGGGNGGCWGCAG-3′ and 5′-GACTACHVGGGTATCTAATCC-3′, respectively) were used to amplify the V3-V4 domain of 16S rRNA as described previously [19]. After the PCR products were purified by AMPure XT beads (Beckman Coulter Genomics, Danvers, MA, USA) and quantified by Qubit (Invitrogen, Carlsbad, CA, USA), the amplicon pools were prepared for sequencing and the size and quantity of the amplicon library were assessed on Agilent 2100 Bioanalyzer (Agilent, Santa Clara, CA, USA) using the Library Quantification Kit for Illumina (Kapa Biosciences, Woburn, MA, USA). The sequencing of libraries was performed on a NovaSeq PE250 platform (Novogene, Beijing, China).

Samples were sequenced on an Illumina NovaSeq platform (LC-Bio, San Diego, CA, USA) according to the manufacturer’s recommendations. Paired-end reads were assigned to samples based on their unique barcode and truncated by cutting off the barcode and primer sequence. Paired-end reads were merged using FLASH version 1.2.7 [20]. Quality filtering on the raw reads was performed under specific filtering conditions to obtain high-quality clean tags according to the fqtrim (v0.94). Chimeric sequences were filtered using Vsearch software (v2.3.4). After dereplication using DADA2, the feature table and feature sequence were obtained. Then, according to SILVA (release 132) classifier, feature abundance was normalized using the relative abundance of each sample. Alpha diversity and beta diversity were calculated by using QIIME2 (Version 1.7.0).

### 2.8. Statistical Analysis

One-way ANOVA was used for statistical analysis of data. The LSD post hoc test was applied for multiple comparisons between groups. For statistical analysis, the software SPSS was used. Data were presented as mean ± standard error, whereby *p* < 0.05 was considered significant. The GraphPad Prism 8.3 software was used to present antioxidant parameters, intestinal morphology, and relative gene expression.

## 3. Results

### 3.1. Growth Performance

The initial BW of chickens did not show significant differences (*p* > 0.05) among the three treatment groups. Broilers in the SL001 group showed significantly higher (*p* < 0.05) BW gains compared to both the control and antibiotic groups and exhibited significantly higher (*p* < 0.05) ADG and ADFI than the control group from day 0 to 42. Furthermore, the F/G ratios of broilers in the SL001 group were significantly lower (*p* < 0.05) than the antibiotic group and displayed a decreasing trend compared with the control group from day 0 to 42 (Table 3). Contrarily, ADG, ADFI, and F/G did not show significant differences among the three treatment groups of broilers (*p* > 0.10) during the first 3 weeks of feeding.

### 3.2. Carcass Traits

The relative weight of the carcass and heart of broilers was significantly higher (*p* < 0.05) in the SL001 group compared to the antibiotic group (Table 4). However, the relative weight of the gizzard was significantly lower (*p* < 0.05) in the SL001 group than in both the control and antibiotic groups. There were no significant differences in the relative weight of breast, liver, abdominal fat, bursa, spleen, stomach, and pancreas among the three experimental groups (*p* > 0.10).

### 3.3. Health-Related Parameters

In order to determine the health status of broiler chickens, three classes of parameters were analyzed in the serum of 42-day broilers including immunological parameters, antioxidant factors, and serum biochemical parameters. For immunological parameters, the results showed that the levels of IL-10, TGF-β, sIgA, immunoglobulin (IgM, IgG, IgA, and IgE) and C4 in the serum of broilers were significantly higher (*p* < 0.001) in the SL001 group than the control group. Contrarily, the serum levels of IL-4, IL-6, and TNF-α in broilers fed on SL001 supplemented diet were significantly lower (*p* < 0.001) than those of the control (Table 5). The levels of IL-6, IL-4, IL-10, TNF-α, TGF-β, IgM, IgA, sIgA, C3 and C4 were not significantly different (*p* > 0.10) between the antibiotic and SL001 groups except the levels of IgG and IgE in serum were significantly decreased (*p* < 0.05) in the SL001 group compared with the antibiotic group. 

For antioxidant factors, the activities of antioxidant enzymes including T-AOC, CAT, and SOD, as well as the content of GSH in broilers with SL001 supplemental diets, were significantly higher (*p* < 0.01) than the control group, with no differences when compared with the antibiotic group (Figure 1). However, the serum MDA concentration of broilers in SL001 and antibiotic treatment groups was significantly lower (*p* < 0.01) than that of the control (Figure 1). For biochemical parameters, our results indicated that SL001 supplementation in diets led to significantly higher levels of ALB and TP (*p* < 0.001) but lower levels UA (*p* < 0.001) compared to the control group at day 42 (Table 6). The activities of AKP, GOT, AMS, and LPS in the SL001 group were significantly higher (*p* < 0.05, respectively) compared to the control group. With daily SL001 supplementation, the serum concentration of HDL at day 42 was significantly higher (*p* < 0.001) than the control group, while the concentrations of TCHO, TG, VLDL and LDL were lower (*p* < 0.001, respectively) than the control group.

### 3.4. Intestinal Morphology 

The effects of *L. reuteri* SL001 on the ileum and jejunum morphology of broilers were shown in Figure 2 and Figure 3. Diets with SL001 supplementation significantly increased the villus height and the ratio of villus height to crypt depth in the ileum of broilers compared to the control (*p* < 0.001, respectively). In addition, a decreased crypt depth (*p* < 0.01) and a higher ratio of villus height to crypt depth (*p* < 0.05) in the jejunum were observed in the SL001 group compared to the control. There was no significant difference in the crypt depth of the ileum and villus height of the jejunum among the three treatment groups (*p* > 0.10, respectively).

### 3.5. Gene Expression Levels in Liver and Jejunum

Figure 4 showed that the relative expression level of genes (*SOD*, *CAT*, *GPX*, and *GHR*) in the liver of broilers were remarkably higher (*p* < 0.05, respectively) in the SL001 group than in the control group and antibiotic group. There were no differences in the expression level of interferon γ (*IFN-γ*), *TGF-β*, and insulin-like growth factor 1 (*IGF-1*) in the liver of broilers between the three treatment groups (*p* > 0.10, respectively). In the jejunum, neither the SL001 diet nor the antibiotic diet affected the expression of detected genes (*IFN-γ*, *TGF-β*, *SOD*, *CAT*, *GPX*, *GHR* and *IGF-1*).

### 3.6. Microbial Diversity and the Relative Abundance of Intestinal Microbiome

The effects of *L. reuteri* SL00L on the alpha diversity of intestinal microbiota of broilers were shown in Table 7. The results showed that dietary SL001 supplementation increased the abundance of intestinal microbiome compared to 592 OTUs in the SL001 group and 567 OTUs in the control group. There were no significant differences in Observed_OTUs, Chao1, Shannon or Simpson parameters of intestinal microbiota between the three treatment groups at day 42 (*p* > 0.10). The SL001 diet resulted in significant changes in the beta diversity of the microbiota of broilers’ cecal contents at day 42 as shown by PCoA (*p* < 0.05) (Figure 5). Analysis of the relative abundance of the intestinal bacterial community showed that Firmicutes and Bacteroidetes were the dominant bacteria, accounting for more than 90% of intestinal bacteria at the phylum level (Table 8). The relative abundance of Firmicutes and Bacteroidetes was not significantly changed between the three treatment groups (*p* > 0.10). However, the SL001 diet significantly increased the relative abundance of Actinobacteria in the cecal contents of broilers (*p* = 0.002). At the genus level, the SL001 diet increased the relative abundance of *Ruminococcaceae_UCG-014* (*p* = 0.058) but decreased the relative abundance of *Faecalibacterium* (*p* = 0.053) and *Parabacteroides* (*p* < 0.05) compared with the control group (Table 9).

## 4. Discussion

Probiotics are defined as live microbial supplements, which are used in the feed to improve the health of animals by balancing the intestinal microbes. *Lactobacillus* bacteria can be used as probiotics as an alternative to antibiotics due to their antimicrobial properties, as well as their beneficial effects on the host [21,22]. Our preliminary studies showed that feeding the *L. reuteri* SL001 strain could improve the structural composition of the mice’s gut bacterial community and play positive roles in adjusting the intestinal bacterial community structure of Alzheimer’s disease model mice [17]. In order to extend the application of this strain, we explored its roles in feeding broiler chickens in the present study. We compared the production performance and carcass traits in broilers fed on a basal diet, a basal diet with zinc bacitracin, and a basal diet with *L. reuteri* SL001. Furthermore, the beneficial effects were demonstrated by an analysis of health-related parameters and intestinal morphology, as well as an analysis of gene expression by qPCR and intestinal microbiome by the 16S rRNA sequencing method.

In the present study, chickens fed on a diet supplemented with SL001 showed greater BW and ADG than did chickens fed a basal diet or chickens fed on antibiotics. Notably, the F/G of broilers in the SL001 group was lower than both the control group and antibiotics groups. These findings were in accordance with previous reports [23,24,25,26], which revealed that incorporating *Lactobacillus* in the basic diet could significantly increase the body weight and reduce the feed/weight ratios of broilers. *Lactobacilli* produce bacteriocins and organic acids, which inhibit the growth and reproduction of pathogenic bacteria in the gut. Meanwhile, lactic acid bacteria can also produce various digestive enzymes, including amylase and protease, which help promote the absorption of nutrients and improve the feed conversion rate [27]. Our results also showed that the effects of SL001 on BW, ADG, ADFI and F/G were more remarkable in 6-week-feeding broilers than in 3-week-feeding broilers. This phenomenon indicated that *L. reuteri* SL001 might take a certain period to colonize and proliferate in the intestinal tract of broilers to exert its biological efficacy, and its efficacy became more significant with increasing intake (Table 3). Meanwhile, a basal diet supplemented with SL001 significantly increased the expression of *GHR* in the liver compared to the control group (Figure 4). *GHR* was reported to play a vital role in animal growth and mutations in the *GHR* gene caused dwarf chicken [28]. The enhanced expression of *GHR* in the liver in the present study revealed that dietary SL001 supplementation might change the expression of developmental genes in broilers fed with SL001.

Immunoglobulins, complement components, and many cytokines are parameters that reflect the immune status of animals because it plays crucial roles in the immune system. *Lactobacillus* affects innate immunity and adaptive immunity after entering the organism, thus contributing to the self-immunity enhancement of the organism [29]. The inflammatory responses were mediated by various cytokines (such as TNF-α and IL-6), whereas the anti-inflammatory responses were mediated by IL-10 and TGF-β, etc. Such inflammatory responses have been found to result in anorexia and reduced growth efficiency in poultry [30,31,32]. Incorporating probiotics into the diet could reduce serum IL-6 levels and increase IL-10 and TGF-β levels in broilers under heat stress [33]. In the present study, our results showed that the application of SL001 significantly reduced the serum levels of IL-6, IL-4, and TNF-α and increased the serum levels of IL-10, TGF-β, IgM, IgG, IgA, IgE, sIgA and C4. These results mainly agreed with the previous studies mentioned above. We deduced that SL001 could inhibit the inflammatory response and improve the growth efficiency of broiler chickens. Furthermore, previous studies had shown that *Lactobacillus* in combination with other probiotics supplementation could increase the levels of IgA and IgM in broilers [34,35,36]. Therefore, *L. reuteri* SL001 could have a positive effect on improving the immunity of broilers. 

Oxidative stress severely affects broiler chickens. Many reports found that the excessive free radical formation, the damage to the antioxidant defense system, and oxidative stress potentially result in the harmful biological condition, which poses severe consequences on the physiological status of poultry [37]. Notably, CAT, GSH, and SOD constitute the body’s antioxidant defense system [38]. Herein, we found that SL001 supplementation significantly improved the activity of CAT, SOD as well as the content of GSH and T-AOC in the serum (Figure 1). Further, compared with the control group and antibiotic groups, we found that the supplementation of SL001 in the diet significantly upregulated the expression of *CAT*, *SOD*, and *GPX* genes in the liver of broilers (Figure 4). These observations implied that SL001 could improve the self-antioxidant defense level and antioxidant capacity of broilers. These results agree with many previous studies [39,40,41]. MDA is a primary degradation product of lipid peroxidation related to oxidative damage in vivo [42]. The addition of SL001 into the feed resulted in a significant reduction in the MDA content of the serum, indicating that SL001 could reduce oxidative stress by inhibiting lipid peroxidation, which was also observed by previous studies [43,44,45]. These results demonstrated that, like or even better than antibiotics, SL001 had a positive role in oxidation resistance, scavenging reactive oxygen species, and stimulating antioxidant capability.

In addition, SL001 supplemental diets significantly increased the serum levels of TP and ALB and the activities of AKP and GOT, whereas UA contents in serum were considerably reduced. TP in the serum mainly comprises albumin and globulin, wherein high albumin contents in the serum, in most cases, reflect a high protein metabolic rate [46]. Uric acid and creatinine are the main products of protein, purine, and muscle tissue metabolism which reflect the health status of the liver [47]. The relatively low levels of UA and Cr could be linked with an improved kidney function of broilers in the SL001 supplementation group. Similar results were reported in previous studies [48] including that the AKP enzyme catalyzes inorganic phosphorus release from multiple lecithin compounds. The present results demonstrated that dietary SL001 supplementation increased the level of AKP in the serum of broilers on day 42, which could be attributed to a lack of phosphorus in the diet or reduced phosphorus absorption in broilers [49,50]. VLDL mainly comprises triglycerides and cholesterol, released by lipase forming ILDL (intermediate LDL) and cholesterol-rich LDL [51]. In the present study, the addition of SL001 into the feed reduced the contents of T-AOC, TG, VLDL and LDL and increased the content of HDL, which was also found in previous studies [52,53].

The intestinal villi provide a high nutrient absorption surface for the intestine, enhancing the maintenance of optimal growth of broilers [27]. Intestinal integrity plays a significant role in preventing the invasion of pathogens in broilers [54,55]. In the present study, we found that supplementation of SL001 in diets significantly affected the villi length of the ileum and the crypt depth of the jejunum of broilers (Figure 2 and Figure 3). Moreover, the ratio of the villi length to the crypt depth suggested that SL001 could promote the development of the intestine and enhance the absorption of nutrients in broilers, which was in agreement with previous studies [56]. Furthermore, our results showed that probiotic SL001 could specifically stimulate the activities of AMS, LPS, and PEP in the serum, implying SL001 is beneficial for the digestion and absorption of nutrients in broilers, which is consistent with the results of weight gain. This can also be confirmed by previous studies that feeding *Lactobacillus* and *Bacillus subtilis* enhanced the activity of digestive enzymes in the gut of broilers [23,57]. 

The composition of gut microbes is critical for gut health, which could in turn affect the systematic health of animals [58]. In this study, SL001 supplementation increased the abundance of intestinal microbiota (not statistically significant) and changed the intestinal microbial structure of broilers. Similarly, studies reported that probiotics supplementation had no effect on the alpha diversity parameters of broilers [59,60]. The increase in intestinal flora diversity may lead to a decrease in flora stability, which is crucial to the health of the host [61]. In addition, it is difficult to determine whether the increase in certain gut microbiota is beneficial to the host or not. Our experimental results also found that after treatment with SL001, the abundance of certain bacteria (such as *Actinobacteria*, *Chloroflexi*, *Acidobacteria*) at the phylum level in cecal contents increased. The possible reason is that these microorganisms have no side effects on the host, or the host immunological system can regulate the microbial community to maintain a balance between the host’s health and the microbial community [62]. Our study showed that *Firmicutes* and *Bacteroidetes* were the dominant bacteria in the gut, which was consistent with previous studies [61,63]. SL001 supplementation in the diet significantly increased the relative abundance of *Actinobacteria* which have a wide range of secondary metabolic functions and produce naturally derived antibiotics, as well as many antifungal compounds [62]. At the genus level, SL001 obviously reduced the relative abundance of *Faecalibacterium* and *Parabacteroides*. *Faecalibacterium* can induce the proliferation of T cells and show anti-inflammatory effects, whilst *Parabacteroides* is associated with weight loss of the host [64]. 

## 5. Conclusions

Overall, dietary *L. reuteri* SL001 supplementation significantly enhanced the growth performance of broilers by strengthening the immunity, relieving antioxidant stress, and improving intestinal morphology and the structure of the intestinal bacterial community of broilers, implying its potential application as an encouraging alternative to antibiotic growth promoters in broiler feeding.

## Figures and Tables

**Figure 1 animals-13-01690-f001:**
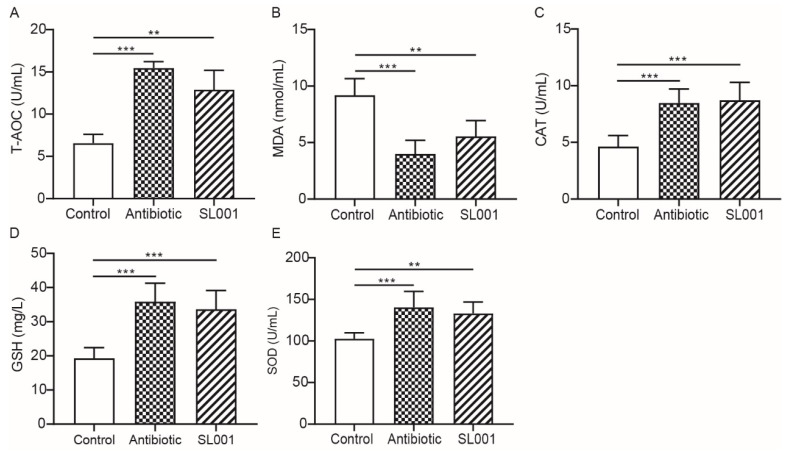
Effect of *L. reuteri* SL001 on antioxidant parameters in the serum of broilers at day 42 (*n* = 6). (**A**) The total antioxidant capacity (T-AOC); (**B**) the malonaldehyde (MDA) content; (**C**) the levels of catalase (CAT); (**D**) the content of reduced glutathione (GSH); (**E**) the levels of superoxide dismutase (SOD). ** *p* < 0.01; *** *p* < 0.001.

**Figure 2 animals-13-01690-f002:**
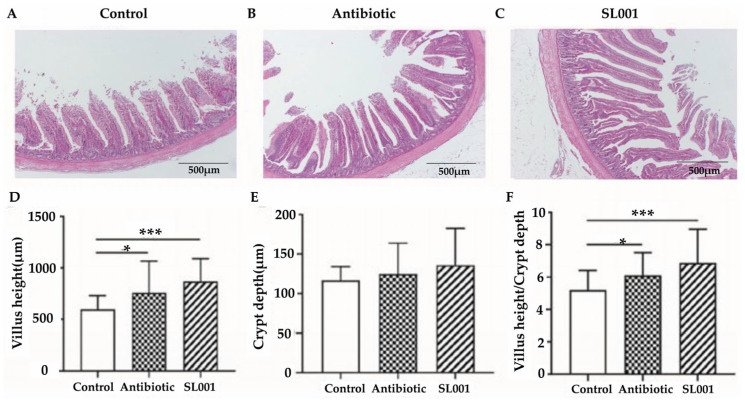
Effects of *L. reuteri* SL001 on ileum morphology of broilers at day 42 (*n* = 6). (**A**–**C**) show the morphology characteristics of ileum from the control group, the antibiotic group, and SL001 group, respectively. (**D**) Comparison of villus height of ileum. (**E**) Comparison of crypt depth of ileum. (**F**) Comparison of the value of villus height/crypt depth. * *p* < 0.05; *** *p* < 0.001.

**Figure 3 animals-13-01690-f003:**
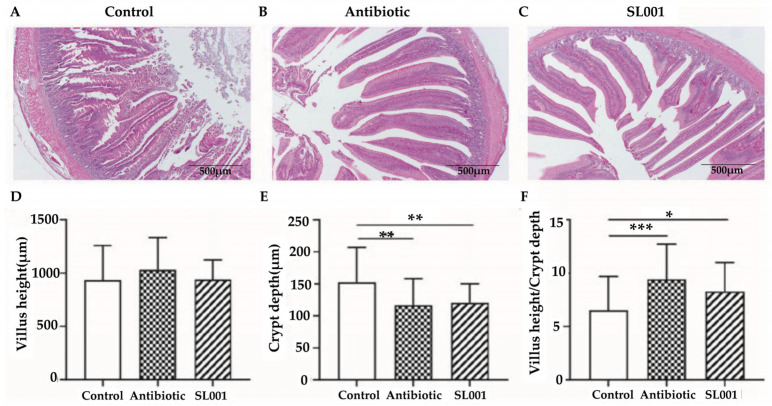
Effects of *L. reuteri* SL001 on jejunum morphology of broilers at day 42 (*n* = 6). (**A**–**C**) show the morphology characteristics of jejunum from the control group, the antibiotic group, and SL001 group, respectively. (**D**) Comparison of villus height of jejunum. (**E**) Comparison of crypt depth of jejunum. (**F**) Comparison of the value of villus height/ crypt depth. * *p* < 0.05; ** *p* < 0.01; *** *p* < 0.001.

**Figure 4 animals-13-01690-f004:**
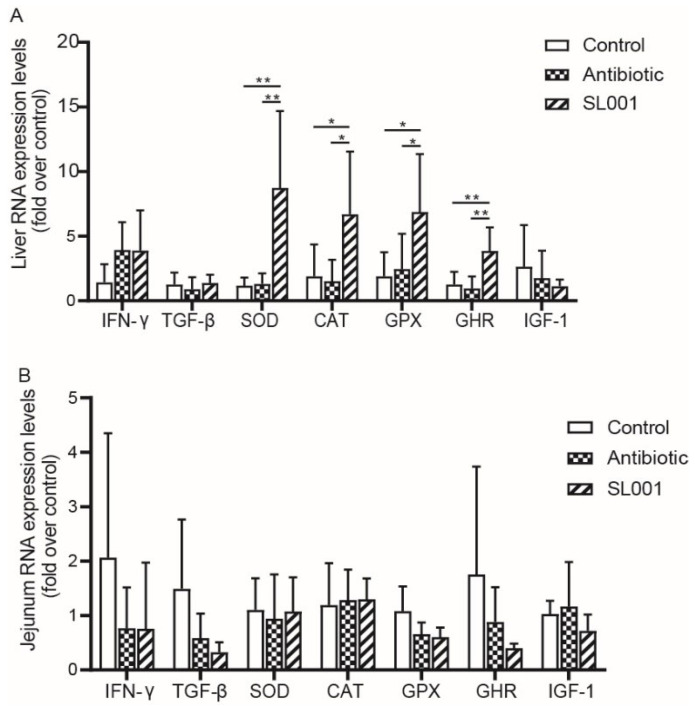
Effects of *L. reuteri* SL001 on gene expression in liver (**A**) and jejunum (**B**) of broilers at day 42 of age (*n* = 6). Abbreviations: IFN-γ, interferon γ; TGF-β, transforming growth factor β; SOD, superoxide dismutase; CAT, catalase; GPX, glutathione peroxidase; GHR, growth hormone receptor; IGF-1, insulin-like growth factor 1. * *p* < 0.05; ** *p* < 0.01.

**Figure 5 animals-13-01690-f005:**
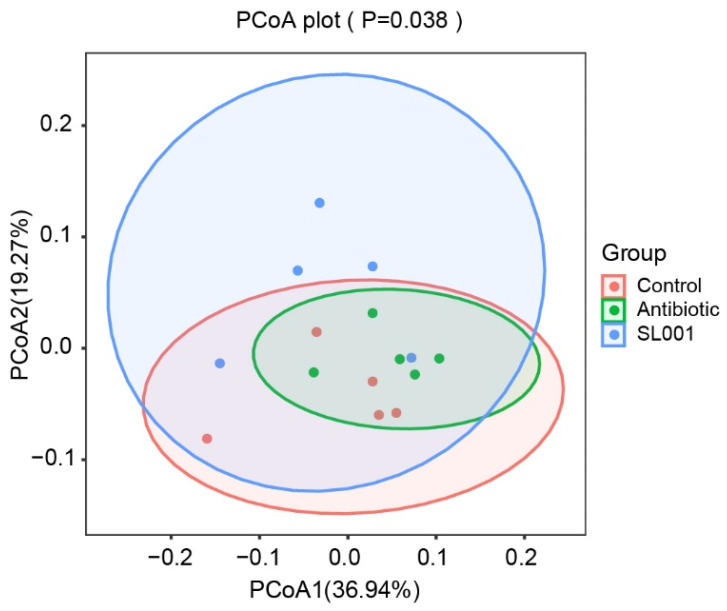
Effects of *L. reuteri* SL001 on intestinal microbial community composition of broilers at day 42 of age (*n* = 5).

**Table 1 animals-13-01690-t001:** Composition and nutrients of basic feed.

Items	1 to 21 d	22 to 42 d
Ingredient (%)		
Corn	58.00	63.00
Soybean (44%)	28.20	24.90
Gluten (60%)	7.00	6.00
Dicalcium phosphate	1.80	1.20
Limestone	1.30	1.30
Oil soybean	2.00	2.00
Salts	0.30	0.30
L-Lys-HCL	0.20	0.20
DL-Met	0.20	0.10
Premix ^1^	1.00	1.00
Total	100.00	100.00
Nutritional ingredient ^2^		
Energy (MJ/kg)	12.60	12.84
Crude protein %	21.99	20.28
Calcium %	0.98	0.84
Total phosphor %	0.72	0.59
Methionine %	0.59	0.47
Methionine + Cysteine %	0.95	0.80
Lysine %	1.14	1.06

^1^ The premix details are provided per kg of premix: vitamin A, 13,000 IU; vitamin D, 13,000 IU; vitamin E, 65 IU; vitamin K, 3.4 mg; vitamin B6, 4.3 mg; vitamin B12, 0.075 mg; pantothenic acid, 37 mg; riboflavin, 6.6 mg; folic acid, 3.7 mg; niacin, 39 mg; thiamine, 1 mg; biotin, 0.23 mg; choline, 0.43 g; Zn, 170 mg; Fe, 140 mg; Mn, 34 mg; Cu, 16 mg; I, 0.29 mg; Se, 0.29 mg. ^2^ Nutrient levels are the calculated values.

**Table 2 animals-13-01690-t002:** Primers used for quantitative real-time PCR.

Gene	Primer Sequence (5′-3′)	Accession Number
*IFN-γ*	Forward: AACGCCAAAGCCTCCTCAAC Reverse: TGAGGTGAAGGTTGCGAGGC	NM_205427.1
*TGF-β*	Forward: CGGGACGGATGAGAAGAAC Reverse: CGGCCCACGTAGTAAATGAT	NM_205454.1
*SOD*	Forward: GCACGGTGGACCAAAAGA Reverse: AACGAGGTCCAGCATTTCC	NM_205064.1
*CAT*	Forward: GGTTCGGTGGGGTTGTCTTT Reverse: CACCAGTGGTCAAGGCATCT	NM_001031215.2
*GPX*	Forward: CAGGAGAACGCCACCAACG Reverse: TCTCAGGAAGGCGAACAGC	NM_001277853.1
*GHR*	Forward: AACACAGATACCCAACAGCC Reverse: AGAAGTCAGTGTTTGTCAGGG	NM_001001293.1
*IGF-1*	Forward: CATTCATTTCTTCTACCTTGGC Reverse: AGCAGCATTCATCCACTATTCC	NM_001004384.2
*β-actin*	Forward: GAGAAATTGTGCGTGACATCA Reverse: CCTGAACCTCTCATTGCCA	NM_205518.1

Abbreviations: IFN-γ, interferon γ; TGF-β, transforming growth factor β; SOD, superoxide dismutase; CAT, catalase; GPX, glutathione peroxidase; GHR, growth hormone receptor; IGF-1, insulin-like growth factor 1.

**Table 3 animals-13-01690-t003:** Productive performance of broilers fed on different diets from 0 to 42 d of age (*n* = 6).

Items	Control	Antibiotic	SL001	*p*-Value
BW (g)				
Initial	47.34 ± 0.39	47.36 ± 0.57	47.74 ± 0.26	0.799
3 weeks	650.26 ± 8.25 ^ab^	635.82 ± 7.56 ^b^	676.42 ± 14.89 ^a^	0.051
6 weeks	1644.41 ± 40.85 ^b^	1607.17 ± 30.39 ^b^	1780.68 ± 34.69 ^a^	0.009
ADG (g/day)				
0–3 weeks	28.71 ± 0.38 ^ab^	28.02 ± 0.36 ^b^	29.94 ± 0.72 ^a^	0.055
4–6 weeks	46.98 ± 1.55 ^b^	46.37 ± 1.45 ^b^	52.78 ± 1.46 ^a^	0.015
0–6 weeks	37.85 ± 0.87 ^b^	37.20 ± 10.732 ^b^	41.36 ± 2.16 ^a^	0.006
ADFI (g/day)				
0–3 weeks	44.14 ± 1.15	43.02 ± 0.97	43.94 ± 1.77	0.823
4–6 weeks	92.06 ± 2.34 ^c^	107.12 ± 1.87 ^a^	100.00 ± 2.36 ^b^	<0.001
0–6 weeks	68.10 ± 1.53 ^b^	75.07 ± 1.08 ^a^	71.97 ± 1.93 ^ab^	0.021
F/G (g/g)				
0–3 weeks	1.54 ± 0.25	1.54 ± 0.53	1.47 ± 0.07	0.624
4–6 weeks	1.96 ± 0.07 ^b^	2.32 ± 0.06 ^a^	1.90 ± 0.08 ^b^	0.002
0–6 weeks	1.80 ± 0.04 ^b^	2.02 ± 0.04 ^a^	1.75 ± 0.06 ^b^	0.004

^a, b, c^ Values with different superscripts within the same row indicate a significant difference at *p* ≤ 0.05. Abbreviations: control, broiler chickens fed on a basic diet; antibiotic, 0.5 g of antibiotics (zinc bacitracin) per kilogram of a basic diet; SL001, 3 × 10^8^ CFU/g basal diet; BW, body weight; ADG, average daily gain; ADFI, average daily feed intake; F/G, ADFI/ADG.

**Table 4 animals-13-01690-t004:** Meat yield, abdominal fat, and organ parameters of broilers from three treatment groups at 42 d (relative to pre-slaughter body weight %) (*n* = 12).

Items (g/g)	Control	Antibiotic	SL001	*p*-Value
Carcass	79.08 ± 0.76 ^ab^	76.36 ± 2.08 ^b^	81.87 ± 0.40 ^a^	0.019
Breast	18.15 ± 0.50	17.49 ± 0.41	18.73 ± 0.51	0.199
Abdominal fat	1.68 ± 0.10	1.62 ± 0.10	1.80 ± 0.15	0.560
Liver	2.42 ± 0.10	2.39 ± 0.11	2.23 ± 0.17	0.580
Gizzard	2.18 ± 0.22 ^a^	2.04 ± 0.10 ^a^	1.44 ± 0.1 ^b^	0.004
Bursa	0.20 ± 0.02	0.21 ± 0.02	0.22 ± 0.02	0.613
Spleen	0.14 ± 0.03	0.10 ± 0.01	0.13 ± 0.01	0.308
Heart	0.45 ± 0.02 ^b^	0.46 ± 0.02 ^b^	0.52 ± 0.03 ^a^	0.050
Stomach	0.41 ± 0.02	0.42 ± 0.02	0.43 ± 0.02	0.915
Pancreas	0.19 ± 0.01	0.19 ± 0.01	0.20 ± 0.01	0.906

^a, b^ Values with different superscripts within the same row indicate a significant difference at *p* < 0.05.

**Table 5 animals-13-01690-t005:** Immunological parameters of 42-day-old broilers from three treatment groups (*n* = 6).

Items	Control	Antibiotic	SL001	*p*-Value
IL-6 (pg/mL)	29.34 ± 1.20 ^a^	16.57 ± 1.44 ^b^	18.58 ± 0.49 ^b^	<0.001
IL-4 (pg/mL)	159.88 ± 2.87 ^a^	103.42 ± 5.00 ^b^	102.49 ± 5.60 ^b^	<0.001
IL-10 (pg/mL)	36.13 ± 2.41 ^b^	63.52 ± 1.52 ^a^	57.72 ± 2.85 ^a^	<0.001
TNF-α (pg/mL)	74.67 ± 3.14 ^a^	50.96 ± 2.94 ^b^	54.85 ± 1.21 ^b^	<0.001
TGF-β (pg/mL)	134.32 ± 4.71 ^b^	195.92 ± 3.65 ^a^	184.12 ± 9.04 ^a^	<0.001
IgM (μg/mL)	895.42 ± 25.63 ^b^	1223.36 ± 29.51 ^a^	1143.87 ± 26.70 ^a^	<0.001
IgA (μg/mL)	514.04 ± 12.70 ^b^	709.08 ± 23.94 ^a^	644.02 ± 5.81 ^a^	<0.001
IgG (μg/mL)	1776.63 ± 78.83 ^c^	2787.04 ± 65.58 ^a^	2339.49 ± 68.89 ^b^	<0.001
IgE (μg/mL)	15.94 ± 0.84 ^c^	22.56 ± 0.67 ^a^	20.18 ± 0.33 ^b^	<0.001
sIgA (ng/mL)	1492.51 ± 63.28 ^b^	2166.70 ± 142.63 ^a^	2172.14 ± 99.97 ^a^	<0.001
C3 (μg/mL)	594.37 ± 12.46 ^b^	769.84 ± 38.57 ^a^	709.93 ± 41.41 ^ab^	0.007
C4 (μg/mL)	136.39 ± 7.08 ^b^	192.21 ± 12.77 ^a^	208.21 ± 8.71 ^a^	<0.001

^a, b, c^ Values with different superscripts within the same row indicate a significant difference at *p* < 0.05. Abbreviations: IL, interleukin; Ig, immunoglobulin; TNF-α, tumor necrosis factor; TGF-β, transforming growth factor; sIgA, secretory immunoglobulin A; C3, complement C3; C4, complement C4.

**Table 6 animals-13-01690-t006:** Serum biochemical parameters of 42-day-old broilers from three treatment groups (*n* = 6).

Items	Control	Antibiotic	SL001	*p*-Value
ALB (g/L)	24.19 ± 1.03 ^b^	37.37 ± 2.24 ^a^	35.31 ± 1.72 ^a^	<0.001
TP (g/L)	49.72 ± 1.25 ^b^	66.89 ± 1.78 ^a^	63.53 ± 1.95 ^a^	<0.001
Cr (μmol/L)	265.71 ± 16.19 ^a^	186.56 ± 6.24 ^b^	218.13 ± 2.52 ^a^	<0.001
UA (μmol/L)	255.97 ± 5.83 ^a^	191.54 ± 8.00 ^b^	208.16 ± 6.49 ^b^	<0.001
AKP (U/L)	60.77 ± 1.49 ^b^	89.14 ± 2.52 ^a^	89.17 ± 3.31 ^a^	<0.001
GPT (U/L)	45.23 ± 6.17 ^b^	57.40 ± 2.15 ^a^	56.81 ± 2.20 ^ab^	0.082
GOT (U/L)	70.17 ± 5.10 ^b^	82.24 ± 2.54 ^ab^	83.49 ± 4.62 ^a^	0.080
AMS (U/dL)	151.00 ± 8.97 ^b^	278.45 ± 19.50 ^a^	254.95 ± 19.53 ^a^	<0.001
LPS (U/L)	593.40 ± 43.50 ^b^	954.49 ± 26.40 ^a^	864.28 ± 36.35 ^a^	<0.001
PEP (U/mL)	46.05 ± 1.25 ^b^	69.02 ± 4.38 ^a^	59.41 ± 5.48 ^ab^	0.005
TCHO (mmol/L)	4.68 ± 0.19 ^a^	2.87 ± 0.18 ^c^	3.44 ± 0.10 ^b^	<0.001
TG (mmol/L)	1.88 ± 0.05 ^a^	1.28 ± 0.06 ^b^	1.37 ± 0.07 ^b^	<0.001
VLDL (mmol/L)	14.01 ± 0.53 ^a^	7.90 ± 0.48 ^b^	7.41 ± 0.50 ^b^	<0.001
LDL (mmol/L)	9.99 ± 0.26 ^a^	6.08 ± 0.39 ^c^	7.71 ± 0.17 ^b^	<0.001
HDL (mg/dL)	110.30 ± 5.79 ^c^	164.16 ± 4.89 ^a^	143.23 ± 4.14 ^b^	<0.001

^a, b, c^ Values with different superscripts within the same row indicate a significant difference at *p* < 0.05. Abbreviations: ALB, albumin; TP, total protein; Cr, creatinine; UA, uric acid; LPS, lipase; GPT, glutamic-pyruvic transaminase; GOT, glutamic-oxalacetic transaminase; AMS, amylase; AKP, alkaline phosphatase; PEP, pepsin; TCHO, total cholesterol; TG, triglyceride; VLDL, very low-density lipoprotein; LDL, low-density lipoprotein; HDL, high-density lipoprotein.

**Table 7 animals-13-01690-t007:** Alpha diversity of intestinal microbiota of broilers from three treatment groups (*n* = 5).

Items	Control	Antibiotic	SL001	*p*-Value
Observed_OTUs	567.20 ± 31.33	581.40 ± 15.92	592.40 ± 20.68	0.755
Chao1	569.23 ± 31.73	583.76 ± 16.39	592.92 ± 20.57	0.781
Shannon	7.00 ± 0.09	7.06 ± 0.10	7.12 ± 0.12	0.739
Simpson	0.978 ± 0.004	0.976 ± 0.002	0.980 ± 0.002	0.679

**Table 8 animals-13-01690-t008:** Effects of *L. reuteri* SL001 on the relative abundance of intestinal bacterial community at the phylum level (*n* = 5).

Items	Control	Antibiotic	SL001	*p*-Value
Firmicutes	71.71 ± 4.18	78.98 ± 2.20	72.53 ± 3.77	0.308
Bacteroidetes	22.16 ± 3.45	16.23 ± 1.94	20.50 ± 3.13	0.364
Proteobacteria	4.35 ± 0.96	3.35 ± 0.48	3.00 ± 0.41	0.357
Tenericutes	0.94 ± 0.22	0.88 ± 0.23	1.57 ± 0.25	0.112
Verrucomicrobia	0.51 ± 0.22	0.30 ± 0.14	1.70 ± 0.72	0.092
Lentisphaerae	0.19 ± 0.12	0.11 ± 0.05	0.07 ± 0.02	0.535
Actinobacteria	0.042 ± 0.008 ^b^	0.078 ± 0.017 ^b^	0.21 ± 0.042 ^a^	0.002
Chloroflexi	0.002 ± 0.002 ^b^	0.000 ± 0.000 ^b^	0.208 ± 0.077 ^a^	0.008
Acidobacteria	0.002 ± 0.002 ^b^	0.000 ± 0.000 ^b^	0.116 ± 0.046 ^a^	0.014
Cyanobacteria	0.076 ± 0.034	0.018 ± 0.010	0.018 ± 0.006	0.109

^a, b^ Values with different superscripts within the same row indicate a significant difference at *p* < 0.05.

**Table 9 animals-13-01690-t009:** Effects of *L. reuteri* SL001 on the relative abundance of intestinal bacterial community at the genus level (*n* = 5).

Genus	Control	Antibiotic	SL001	*p*-Value
*Faecalibacterium*	13.02 ± 1.91 ^a^	13.12 ± 1.19 ^a^	7.57 ± 1.72 ^b^	0.053
*Ruminococcaceae_UCG-014*	6.40 ± 0.67 ^b^	7.53 ± 0.92 ^ab^	10.27 ± 1.40 ^a^	0.058
*Alistipes*	6.18 ± 2.08	8.48 ± 1.63	9.09 ± 1.60	0.497
*Ruminococcaceae_UCG-005*	5.08 ± 0.73	6.23 ± 1.67	6.02 ± 1.72	0.839
*Parabacteroides*	9.85 ± 1.79 ^a^	3.30 ± 1.53 ^b^	3.98 ± 2.01 ^b^	0.046
*Bacteroides*	4.85 ± 1.46	3.28 ± 0.72	6.22 ± 1.81	0.367
*Lachnospiraceae_unclassified*	4.21 ± 0.64	4.68 ± 0.44	3.75 ± 0.51	0.493
*Clostridiales_vadinBB60_group_unclassified*	3.07 ± 0.42 ^b^	3.58 ± 0.40 ^ab^	5.62 ± 1.26 ^a^	0.100
*Intestinimonas*	4.92 ± 0.63	3.74 ± 0.88	3.61 ± 0.71	0.424
*Lactobacillus*	2.76 ± 0.63 ^b^	4.96 ± 0.74 ^a^	2.76 ± 0.28 ^b^	0.031

^a, b^ Values with different superscripts within the same row indicate a significant difference at *p* < 0.05.

## Data Availability

Raw data are held by the author and may be available upon request.

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
