# Peer review of "Dietary Lactobacillus reuteri SL001 Improves Growth Performance, Health-Related Parameters, Intestinal Morphology and Microbiota of Broiler Chickens"

_animals, 2023, doi:10.3390/ani13101690_

Round 1

Reviewer 1 Report

This manuscript studies the potential use of a probiotic (Lactobacillus reuteri SL001) to promote the health status of broiler chickens. The effects were investigated on growth performance, carcass traits, serum biochemical parameters, gene expression, gut morphology, and microbiota analysis by comparing with a growth promotion antibiotic. The research is interesting and applicable. I can say this paper is suitable for publication with minor changes. I would like to recommend you the following suggestions.

General comments:

The manuscript is well-structured and clearly written. However, a minor spell and style checker is required. The research design is appropriate with the proposed objectives, the methods are adequately described. The results are clearly presented in tables and figures. The discussion can be improved.

Specific comments are discussed below:

Line 23. The abstract should be a total of about 200 words maximum.

Line 106. The authors described that “In the SL001 treatment group, chickens were fed on basal diet containing 3 × 108 CFU/g SL001 bacterial suspension”, please revise if it is correct.

Lines 134 and 157. How many animals were analyzed?

Line 177. Should say 16S rRNA instead of 16S rDNA sequencing and data analysis.

Line 199. Does all the data meet the assumptions for one way anova? Please revise if it is correct.

Line 248. Reduced glutathione (GSH) is not an enzyme. Please, describe the results as content instead of activity in text. As far I can see, caption of figure 1 is correct.

Lines 297 - 299. Revise the sentence. It seems that the term "were observed" is unnecessary.

Line 428. The authors need to include more information to improve the discussion of microbiota results.

Line 464. Check the references for correct formatting.

Author Response

Thank you very much for your suggestion.

Point 1: Line 23. The abstract should be a total of about 200 words maximum.

Response 1: Thanks for the suggestion. We have tried our best to shorten the Abstract to about 200 words.

Point 2: Line 106. The authors described that “In the SL001 treatment group, chickens were fed on basal diet containing 3 × 108 CFU/g SL001 bacterial suspension”, please revise if it is correct.

Response 2:  This sentence was revised to “In the SL001 treatment group, chickens were fed on basal diet containing 3 × 108 CFU/g SL001 bacteria”. We have confirmed again that the concentration of SL001 bacteria added to basal diet at the final concentration of 3 × 108 CFU/g.

Point 3: Lines 134 and 157. How many animals were analyzed?

Response 3: Line 134, For part of Measurements of health-related parameters, 12 healthy broiler chickens were randomly selected from each treatment group (120 chickens), sacrificed, and used for parameters measurement . Line 157, for part of Total RNA isolation and quantitative real-time PCR analysis, 7 independent biological repeats were performed by using materials from 7 broiler chickens.

Point 4: Line 177. Should say 16S rRNA instead of 16S rDNA sequencing and data analysis.

Response 4:  The statement of “16S rDNA sequencing” is correct. It was that the gene of 16S rDNA was sequenced.

Point 5: Line 199. Does all the data meet the assumptions for one way anova? Please revise if it is correct.

Response 5: Yes. Data on growth performance, serum-biochemical indices, intestinal morphology, and gene expression in the liver and jejunum meet the assumptions for one-way ANOVA.

Point 6: Line 248. Reduced glutathione (GSH) is not an enzyme. Please, describe the results as content instead of activity in text. As far I can see, caption of figure 1 is correct.

Response 6: Thank you very much for your suggestion. We have corrected the description of GSH as content instead of activity.

Point 7: Lines 297 - 299. Revise the sentence. It seems that the term "were observed" is unnecessary.

Response 7: Thanks for your suggestion. We have deleted the term “were observed”.

Point 8: Line 428. The authors need to include more information to improve the discussion of microbiota results.

Response 8: We have improved the discussion of microbiota results.

Point 9: Line 464. Check the references for correct formatting.

Response 9: Thanks. We have corrected the format of references.

Reviewer 2 Report

Comments to the Author:

This study was conducted to study the effects of probiotic SL001 on growth performance, health-related parameters, intestinal morphology and microbiota of broilers. There are some problems:

Major comments:

1.     Too many grammar mistakes. The language does not meet the requirements of publication and needs improvement by professional company or local speaker.

2.     There has been considerable literature on the effects of probiotics as antibiotic substitution on growth, parameters and gut microbiota of broilers, the findings of current study are not particularly innovative.

3.     line40: The feces and urine of chickens are mixed together, how did the authors separate the feces of chickens?

4.     Only genus and species names of bacteria should be written in italic, please change the phylum names.

5.     Table1. Confirm it. Air dry basis or as fed basis?

6.     Table3. n=120?

7.     line155: jejunum tissues or jejunum mucosa?

8.     line175: do the authors mean cecal content?

9.     line208-210: “at day 42”. The expression is not accurate. The performance is the result of a stage rather than a point in time. “at day 42 ,……” is day 0 to 42 or day 21 to 42? Please describe clearly.

10.  Please give the full names of the abbreviations in the table note.

11.  line 309-310: as shown in Figure 5, the P value for β diversity was 0.038.

12.  Discussion: lack of the discussion of the effects of antibiotic on broilers.

13.  Discussion: The potential modes of actions of SL001 on immunity were not discussed properly.

14.  Discussion: As for the antioxidation, antibiotic also increased the CAT, GSH and SOD, and decreased the MDA in serum, like SL001. However, the authors did not discussed this.

Minor comments:

1.     title: Dietary Lactobacillus reuteri SL001 improves…

2.     line17: In the present study

3.     line19: Our results showed…

4.     line19-20: in diets promoted…strengthened…improved…

5.     line24: “360 1-day-old…”Arabic numerals cannot be placed at the beginning of a sentence.

6.     Abstract: write the full names of abbreviations, such as AA, BW, VLDL, LDL.

7.     line39: “did not change too much in gut of broilers supplemented with SL001. “,  what does “did not change too much” mean? How much change is “much change”?

8.     line40: change P=0.002 to P<0.01.

9.     line60-61: improves the diversity…

10.  line67: …inhibit pathogens.

11.  line68: Lactobacillus plantarum

12.  line74: rewrite this sentence

13.  line 74: “the lactobacillus has no…”, what is“the lactobacillus” refers to? SL001?

14.  line97: “day1”do you mean one day old?

15.  lines100-102: rewrite the sentence

16.  lines124,147: How were the 12 or 6 broilers selected? Two or one from each replicate?

17.  line142: pepsin

18.  line205:the full name only needs to be written when the abbreviation first appears.

19.  line205: P=0.799 revised to P>0.05

20.  change figure1 into table as table 5 and 6.

21.  line316: feces

22.  line339: in the present study

23.  line356-358: any references to support?

Author Response

Thank you very much for your suggestions.

Major comments:

Point 1: Too many grammar mistakes. The language does not meet the requirements of publication and needs improvement by professional company or local speaker.

Response 1: Thank you for your suggestion. Actually, we had asked a professional English editor to improve the language of our manuscript before we submitted it. We had checked and corrected the grammar mistakes again.

Point 2: There has been considerable literature on the effects of probiotics as antibiotic substitution on growth, parameters and gut microbiota of broilers, the findings of current study are not particularly innovative.

Response 2: Yes. You are right. There has been many literature on the effects of probiotics as antibiotic substitution on growth, parameters and gut microbiota of broilers. But our bacteria is special that it is able to grow in vitro with cholesterol as a carbon source.

Point 3:  line40: The feces and urine of chickens are mixed together, how did the authors separate the feces of chickens?

Response 3: We did not separate the feces and urine of chickens. The samples for microbiome 16S rDNA sequencing were collected directly from dissected cecal. We have revised “cecal feces” to “cecal contents”.

Point 4:  Only genus and species names of bacteria should be written in italic, please change the phylum names.

Response 4: Thanks a lot. We have corrected this in the manuscript.

Point 5:  Table1. Confirm it. Air dry basis or as fed basis?

Response 5: Should be fed basis.

Point 6: Table3. n=120?

Response 6: Yes, n=120, for each of the treatment group of Control, Antibiotic and SL001.

Point 7:  line155: jejunum tissues or jejunum mucosa?

Response 7: Jejunum tissues.

Point 8:  line175: do the authors mean cecal content?

Response 8: Thanks. We have revised “cecal feces” to “cecal content” throughout the manuscript.

Point 9:    line208-210: “at day 42”. The expression is not accurate. The performance is the result of a stage rather than a point in time. “at day 42 ,……” is day 0 to 42 or day 21 to 42? Please describe clearly.

Response 9: Thanks for your suggestion. We have corrected.

Point 10:   Please give the full names of the abbreviations in the table note.

Response 10: Thanks. We have give all the full names of the abbreviations in the table note.

Point 11:  line 309-310: as shown in Figure 5, the P value for β diversity was 0.038.

Response 11: We have revised the sentence about β diversity (Figure 5).

Point 12:  Discussion: lack of the discussion of the effects of antibiotic on broilers.

Response 12: We have add the discussion of the effects of antibiotic on broilers.

Point 13:  Discussion: The potential modes of actions of SL001 on immunity were not discussed properly.

Response 13: We have revised.

Point 14:  Discussion: As for the antioxidation, antibiotic also increased the CAT, GSH and SOD, and decreased the MDA in serum, like SL001. However, the authors did not discussed this.

Response 14: We have add the discussion of the effects of antibiotic on broilers.

Minor comments:

Point 1: title: Dietary Lactobacillus reuteri SL001 improves…

Response 1: Thanks. We have revised.

Point 2:  line17: In the present study

Response 2: Revised.

Point 3:  line19: Our results showed…

Response 3: Revised.

Point 4:  line19-20: in diets promoted…strengthened…improved…

Response 4: Revised.

Point 5:  line24: “360 1-day-old…”Arabic numerals cannot be placed at the beginning of a sentence.

Response 5: Revised.

Point 6:   Abstract: write the full names of abbreviations, such as AA, BW, VLDL, LDL.

Response 6: Sorry. We can not do that because word count of the abstract has exceeded the limit. We had gave the full names of abbreviations in the main text.

Point 7:  line39: “did not change too much in gut of broilers supplemented with SL001. “,  what does “did not change too much” mean? How much change is “much change”?

Response 7: Thank you very much for your suggestion. We have revised the sentence.

Point 8:   line40: change P=0.002 to P<0.01.

Response 8: Revised.

Point 9:  line60-61: improves the diversity…

Response 9: Revised.

Point 10: line67: …inhibit pathogens.

Response 10: Revised.

Point 11.  line68: Lactobacillus plantarum

Response 11: Revised.

Point 12.  line74: rewrite this sentence

Response 12: Revised.

Point 13.  line 74: “the lactobacillus has no…”, what is“the lactobacillus” refers to? SL001?

Response 13: SL001. We have revised.

Point 14.  line97: “day1”do you mean one day old?

Response 14: Yes. We have revised.

Point 15.  lines100-102: rewrite the sentence

Response 15: We have revised.

Point 16.  lines124,147: How were the 12 or 6 broilers selected? Two or one from each replicate?

Response 16: Yes. Two or one broilers were selected randomly from each replicate.

Point 17.  line142: pepsin

Response 17: Revised.

Point 18.  line205:the full name only needs to be written when the abbreviation first appears.

Response 18: Revised.

Point 19.   line205: P=0.799 revised to P>0.05

Response 19: Revised.

Point 20.  change figure1 into table as table 5 and 6.

Response 20: Sorry. There were too many tables in this manuscript. We don’t think it is necessary to change figure 1 into table.

Point 21.   line316: feces

Response 21: Revised.

Point 22.  line339: in the present study

Response 22: Revised.

Point 23.   line356-358: any references to support?

Response 23: References 25-28.

Reviewer 3 Report

see attachment! 

Author Response

Thank you very much for your suggestions.

Point 1: Line 26, “20 birds/replicate” which experimental design was followed?

Response 1:  Our experimental design was followed reference “Mohamed TM, Sun W, Bumbie GZ, et al. Feeding Bacillus subtilis ATCC19659 to Broiler Chickens Enhances Growth Performance and Immune Function by Modulating Intestinal Morphology and Cecum Microbiota. Front Microbiol. 2022;12:798350. doi:10.3389/fmicb.2021.798350”.

Point 2: Line 33, “P < 0.05,” less and equal to.

Response 2: We have corrected.

Point 3: Lines 98, “2.2 Study design and animal management”, which experimental design was followed?

Response 3: As we answered to Point 1, our experimental design was followed reference “Mohamed TM, Sun W, Bumbie GZ, et al. Feeding Bacillus subtilis ATCC19659 to Broiler Chickens Enhances Growth Performance and Immune Function by Modulating Intestinal Morphology and Cecum Microbiota. Front Microbiol. 2022;12:798350. doi:10.3389/fmicb.2021.798350”.

Point 4: Line 199, “2.8 Statistical analysis”, add mathematical model for better illustration of data analysis. 

Response 4: Sorry. We can not do that.

Point 5: Line 201, ”multiple comparisons between groups”, for statistical analysis which software was used?

Response 5: For statistical analysis, the software SPSS was used.

Point 6: Line 205, “3. Results”, in result section add actual p-value of each parameter.

Response 6: Do you mean that we should give the actual p-value as “P=X” instead of “P < 0.05”? Unfortunately, the second reviewer gave different suggestions.

Point 7: Lines 218, “P < 0.05”, less and equal to

Response 7: We have revised.

Point 8: Line 335, “4. Discussion”, this section needs major attention, add logical reasoning of each result before discuss with previous studies. 

Response 8: We have revised.

Round 2

Reviewer 2 Report

1.  The novelty of the study requires better distinction in the Introduction.

2. Table 3: I think the growth performance in this experiment was measured on replication basis, so the n should be 6, instead of 120.

3. The growth performance is the result of a stage (e.g. from day 0 to day 42), but the other parameters were only tested at day 42. Thus, the description “from day 0 to 42”in line 251, 254, 259, 280, 286, 300, 310. 311…were not correct.

4. Compared to the Control group, the antibiotic treatment also had some significant effects on the tested parameters. However, in the discussion section, the authors only compared the SL001 with Control and Antibiotic. The authors should also discussion why the antibiotic treatment led to the increased antioxidant capacity, improved intestinal morphology and altered gut microbiota…

Author Response

Point 1.  The novelty of the study requires better distinction in the Introduction.

Response 1: Lactobacillus reuteri SL001 was screened and selected by using cholesterol as the only carbon source. We have provided this information in the Introduction. Our previous results also showed dietary SL001 affected bile production in mice,  and we don’t think it is reasonable to add this information in the Introduction since it is unpublished results.

Point 2. Table 3: I think the growth performance in this experiment was measured on replication basis, so the n should be 6, instead of 120.

Response 2: Thanks a lot. We now understand that your suggestion is correct, and we have revised it in the manuscript.

Point 3. The growth performance is the result of a stage (e.g. from day 0 to day 42), but the other parameters were only tested at day 42. Thus, the description “from day 0 to 42”in line 251, 254, 259, 280, 286, 300, 310. 311…were not correct.

Response 3: We have revised. Thank you very much for your patience to help us improve our manuscript.

Point 4. Compared to the Control group, the antibiotic treatment also had some significant effects on the tested parameters. However, in the discussion section, the authors only compared the SL001 with Control and Antibiotic. The authors should also discussion why the antibiotic treatment led to the increased antioxidant capacity, improved intestinal morphology and altered gut microbiota…

Response 4: We are sorry for lack of discussion on zinc bacitracin (Antibiotic) treatment group in Discussion section. Our results showed that zinc bacitracin treatment group exhibited positive effects on broiler growth performance, oxidative stress resistance, immunity, and intestinal morphology. These results were in accordance with many previous reports (Li J,2022; Waldroup PW, 1990). This research mainly focused on the effects of L. reuteri SL001 and We don’t think it is necessary to discuss why the antibiotic treatment led to the increased antioxidant capacity, improved intestinal morphology and altered gut microbiota. Frankly speaking, we don’t know how to do.

Li J, Xiao Y, Fan Q, Yang H, Yang C, Zhang G, Chen S. Dietary Bacitracin Methylene Disalicylate Improves Growth Performance by Mediating the Gut Microbiota in Broilers. Antibiotics (Basel). 2022, 11(6):818. doi: 10.3390/antibiotics11060818.

Waldroup PW, Izat AL, Primo RA, Twining PF, Herbert JA, Trammell JH, Fell RV, Crawford JS. The effect of zinc bacitracin and roxarsone on performance of broiler chickens when fed in combination with narasin. Poult Sci. 1990, 69(6):898-901. doi: 10.3382/ps.0690898.
